# Proanthocyanidins Should Be a Candidate in the Treatment of Cancer, Cardiovascular Diseases and Lipid Metabolic Disorder

**DOI:** 10.3390/molecules25245971

**Published:** 2020-12-16

**Authors:** Tsz Ki Wang, Shaoting Xu, Shuang Li, Yunjian Zhang

**Affiliations:** Department of Thyroid, Breast Surgery, The First Affiliated Hospital, Sun Yat-Sen University, Guangzhou 510080, China; jadeziawang@hotmail.com (T.K.W.); xushaotingst@163.com (S.X.); lish326@mail.sysu.edu.cn (S.L.)

**Keywords:** proanthocyanidins, cancer, cardiovascular disease, lipid metabolism, antioxidant, anti-inflammation, signal pathway

## Abstract

The conventional view of using medicines as routine treatment of an intractable disease is being challenged in the face of extensive and growing evidence that flavonoids in foods, especially proanthocyanidins (PAs), can participate in tackling fatal diseases like cancer, cardiovascular and lipid metabolic diseases, both as a precautionary measure or as a dietary treatment. Although medical treatment with medicines will remain necessary in some cases, at least in the short term, PAs’ function as antioxidant, anti-inflammatory drugs, signal pathway regulators remain critical in many diseases. This review article demonstrates the physical and biological properties of PAs, summarizes the health benefits of PAs found by researchers previously, and shows the possibility and importance of being a dietary treatment substance.

## 1. Introduction

Dietary flavonoids have been reported to possess substantial anticarcinogenic and health care activities because of their antioxidant and anti-inflammatory properties. Proanthocyanidins (PAs) are considered as the key ingredient, and are abundantly available in various parts of plants, such as the fruits, barks and plant seeds. Their modes of action were evaluated through a number of studies which showed their potential role as therapeutics of cancer, cardiovascular diseases and lipid metabolic disorder, etc. Here we summarize the structure, isolation, characterization, synthesis and health benefits of PAs. The modulation of various molecular targets by PAs in previous studies suggests their importance, contribution and mechanism of action in the prevention of cancers, cardiovascular diseases and lipid metabolic disorder.

## 2. Properties and Isolation of PAs

### 2.1. Structure

Flavonoids consist of 15 carbon atoms, forming structures with two aromatic rings connected by a three-carbon bridge. Dietary flavonoids can be divided into six major groups (Table 1): flavonols, flavones, flavan-3-ols, anthocyanidins, flavanones, and isoflavones [1]. The components of PAs are the monomers (+)-catechins (CC) and the isomer (−)-epicatechins (EC) in flavan-3-ols. The (+)-catechins and (−)-epicatechins connect with each other or themselves via oxidative coupling between the C4 of the heterocyclic ring and the C8 of the adjacent aromatic ring, which form proanthocyanidins B1–B4. When the C4 is connected to the C6, they form B5–B8. Type-A PAs differ in an ether bond created between the C2 from (−)-epicatechins and the C7 from the former or (+)-catechins [2]. Researchers [3] classify PAs according to the degree of polymerization (DP). Depending on the number of subunits of flavan-3-ols from 1 to 4, PAs are divided into four classes: monomers, dimers, trimers, and tetramers. PAs with DP > 4 are classified as oligomers, and DP > 10 are polymers [4].

### 2.2. Isolation and Metabolism of PAs

PAs are found in nearly half of dietary fruits and plants, with high concentrations in the skin of fruits and leaves, even in grape seeds [5]. The methods and efficiency of PA isolation have improved with the optimization of methods and the progress of instruments. Nonetheless, the preparation procedures of proanthocyanidins from different parts of various plants or fruits are dissimilar, but predominantly contain three steps: pre-treatment, extraction, and purification [6]. Pre-treatment involves grinding fruits and plants rich in PAs into a powder first. According to the conventional extraction methods, the ground powder is then extracted by organic solvents with acetone/water (70:30, *v*/*v*) as the optimal solvent in recent years and concentrated to remove the solvent [7,8,9,10]. However, long extraction time and excessive consumption of organic solvents limit the efficiency improvement and popularization of these techniques, simultaneously with low reproducibility. Various advanced methods, such as ultrasound-assisted extraction (UAE), enzymatic extraction, ultra-high-pressure-assisted extraction, have also been reported for proanthocyanidin extraction [11,12,13,14,15]. So far, no uniform standard has been established. Based on the distinct solubility in eluent or adsorption capacity of resin, thin-layer chromatography (TLC) and Sephadex LH-20 column are currently applied to elute the crude extract into fractions rich in PAs [16,17]. Zhang et al. demonstrated that using high-speed counter-current chromatography (HSCCC) alone, or combined with semipreparative High Performance Liquid Chromatography (HPLC) could obtain PAs in different DP levels with high purity [18]. Next, the composition, structure and DP of each fraction could be analyzed by matrix-assisted laser desorption/ionization-time of flight mass spectrometry (MALDI-TOFMS), and even 13C-nuclear magnetic resonance (^13^C-NMR) analysis [19].

The DP of PAs determines their bioavailability and subseq uent pharmacology prominently. Under oral administration, the carbon bonds connecting monomers are broken due to the low pH value of gastric juice, leading to PA degradation [4]. An in vitro colonic carcinoma (caco-2) cell study demonstrated that CC, dimers and trimers could be absorbed through the intestinal epithelium [20]. Therefore, monomers and oligomers are preferentially absorbed by enterocytes for glucuronidation in the small intestine, but unabsorbed PAs with DP > 4 are fermented by colonic microbiota [9,21]. A study in mice has revealed that these proanthocyanidins with a high degree of polymerization could affect the activity of digestive enzymes and the absorption of other nutrients, suggesting that in addition to the large molecular weight and excessive OH groups, this might be another reason why polymeric proanthocyanidins were difficult to absorb [22]. Deeper metabolites sulfurated and methylated in liver by phase II enzymes then reach the target tissues for pharmacological action and are ultimately excreted through the urinary system and digestive system [21,23]. It has been reported that monomers, dimers, and their metabolites, such as catechin-glucuronides, epicatechin-glucuronides, methyl-catechin-glucuronides and methyl-epicatechin-glucuronides were detected in the serum of rats treating with grape seed proanthocyanidin extracts (GSPE) [24]. In a randomized cross-over study in humans, after ingesting a single dose of EC and PA B1 of 1 mg per kg body weight, respectively, the maximum plasma concentration of EC and its metabolites ranged from 89~190 ng/mL, while the maximum levels of PA B1 and its metabolites was 2.0 ng/mL [25]. This wide range of metabolites in blood subsequently reached the target organs and precisely exerted pharmacological effects. Furthermore, the distribution of PA metabolites could be observed in rats that ingested the hazelnut skin extract (5 g per kg body weight) whose essential component was 6.3 ± 0.54 μmol·g^−1^ CC, 2.4 ± 0.13 μmol·g^−1^ EC, 15 ± 1.3 μmol·g^−1^ PA dimers and 4.9 ± 0.32 μmol·g^−1^ PA trimers [26]. It revealed that lung, kidney, thymus, testicle and spleen were the places with a high concentration of the phase II metabolites of PAs, but not dimers and trimers. As for excretion, Zhang et al. showed that 58–78% PAs were excreted via urine and faeces, which indicated a potential low bioavailability in prototype drugs [4]. Nonetheless, supported by molecule modification technology or packaged with nanoparticles, PAs should immensely fulfil pharmacological effects.

### 2.3. Biological Properties of PAs

Well-known multifunctional substances in biological science, PAs, from low molecular weight to the high one, give significant advantages for our health [27]. Studies on PAs have been widely carried out and achieved remarkable results based on cell and animal models. However, few clinical trials have been conducted, which limits its application as a diet-therapeutic ingredient or as a clinical candidate. Estimated data from the United States Department of Agriculture (USDA)’s Continuing Survey of Food Intake by Individuals (CSFII) show that the mean daily intake of PAs in 1994–1996 was 53.6 mg/person/day excluding monomers and 57.7 mg/person/day including monomers, which is inadequate for its health benefits [28,29]. Clinical trials of PAs are still insufficient to provide substantial evidence for the recommendation of diet therapy using PAs. Such clinical trials were often based on the use of various forms of food, like complex food intake adhering to dietary guidelines, fruits, GSPE or even extracts from French Maritime Pine bark (FMPB), which were then either hard to evaluate the effectiveness or not commercially available, in order to provide a strong demonstration for the benefits of PAs as a diet-therapeutic ingredient [30,31,32,33,34]. Under these clinical trials, PAs themselves were insufficiently proved by undergoing such process for their biological properties. It might be caused by the low absorption of PAs individually and the dose used in the trial (only 130 mg) [29]. The complex form of oligomeric PAs is necessary for the following reasons: (1) As the main ingredient in such complexes like GSPE, PAs are about 39–90%, which is sufficient to have biological properties [21,27,31,35,36,37]. (2) Other ingredients have supplementary effects. For example, (−)-epicatechin and soy phospholipids are used to improve the pressure-regulating effect and enhance bioavailability, respectively [27,32,33,36]. The selection of the type of PA mixture still needs to be considered as PAs of different origins have different compositions [38]. So far plant or fruit extracts rich in a variety of oligomeric PAs, are the primary interventions in vitro and in vivo studies, and the purity of the PAs is up to 90%, which ensures the effectiveness of studies and the verification of pharmacological effects. Research of individual PAs is also emerging gradually, which facilitates an extended cognition and accurately defines the real active substances targeting the mechanisms of specific diseases. Up to now, however, only 4 of 40 completed human clinical trials on PAs or GSPE focused on individual PAs, leaving an essential gap in this field (www.clinicaltrials.gov). Thus, GSPE and leucoselect phytosome (LP) might be better candidates for PA employment as they are commonly used in different experiments, can be orally taken, and have a mature product such as Gravinol [32,33,39]. They are both standardized GSPE complexes (dimer B-3 and trimer C-2) that can be easily absorbed [33].

One of its pharmacological effects, antioxidant activity, is significant in preventing cardiovascular diseases or metabolic syndromes [40]. Another systematic effect is mainly about glycometabolism and lipid metabolism, respectively. Previously, some researchers found that proanthocyanidin extracts can protect β cells in the pancreas by dealing with oxidative stress, enhancing the sensitivity and secretion of insulin, or even affecting some enzyme activities in the metabolic process [41,42,43]. For the past decade, clinical trials based on PA usage were mainly for the cardiovascular and lipid metabolic effect. A randomized, double-blind placebo-controlled clinical trial for eight weeks with 70 patients using grape seed extract (GSE) 200 mg or placebo were conducted to indicate that GSE increases paraoxonase (PON) activity mostly through increasing HDL-C and apo-AI levels in moderate hyperlipidemia (MMH) patients [44]. Another randomized, double-blinded study using additional PA-containing supplement, Aterofisiol, based on acetylsalicylic acid 100 mg for 25 days could ameliorate the amount of cholesterol and lipid in atherosclerosis plaque [45]. An epidemiology study in 2020 also stated that, by consuming two apples a day (about 200 mg PAs per day), PAs could lower correlated lipid index including serum cholesterol, LDL oxidation, raise HDL cholesterol, and restrict further processes happening in atherosclerosis by activation of endothelial nitric oxide synthase, prevention of platelet aggregation, and blockage of inflammatory responses [31]. These trials come up with the strength of PAs as beneficial in both prophylactic and clinical medicine. However, a phase II, placebo-controlled, randomized trial involving early breast cancer patients demonstrated that consuming GSPE (300 mg per day) for six months did not ameliorate the tissue hardness due to radiotherapy [46]. On the other hand, trials within lung cancer by using LP could reduce cancer index serum miR-19a, -19b, and -106b and an average of 55% bronchial Ki-67 in heavy active, former smokers. Phase I lung cancer chemoprevention trial with LP was shown to be succeeded by Mao, J. T. et al. [32,33]. Conflicting results from different clinical trials might be due to the different distribution of PAs in different organs and the concentration of metabolites in target tissues did not meet the pharmacological effect requirement. PAs proved their potential usage in cancers, cardiovascular diseases, endocrine diseases, osteoarticular diseases, urinary diseases and even mental disorders. Further usage in other systems still being explored [39,47,48,49,50,51,52]. In this review, we principally describe the health benefits of PAs in fatal diseases such as cancers, cardiovascular diseases, and lipid metabolism disorders based on in vitro and in vivo studies.

## 3. Health Benefits of PAs

### 3.1. Anticarcinogenic Activities

In recent years, increasing studies have found that PAs exert an anticancer function, including the reduced risk and treatment of cancer [53,54]. Being natural anticancer agents, PAs disturbed multiple pathways associated with the growth and progression of various cancers, such as breast cancers, colorectal cancers, and prostatic cancers [54,55]. Here we summarize the dominating mechanisms of PAs in these common cancers (Figure 1).

#### 3.1.1. Angiogenesis

Angiogenesis is a critical step in tumour growth and metastasis. Proangiogenic factors secreted by cancer cells promote new blood vessels, which provide nutrients for tumour growth. Among proangiogenic factors, vascular endothelial growth factor (VEGF) contributes to forming new blood vessels, while angiopoietin 1 (Ang1) takes charge of maturation and stabilization [56,57]. GSPE have been shown to inhibit these molecules from binding to receptors on endothelial cell surfaces in colon cancer [58]. In breast cancer, GSPE suppresses endothelial cells’ proliferation and migration by interfering with the VEGF-MAPK pathway both in vitro and in vivo [59]. The inhibitory effect of PAs on the expression of VEGF and hypoxia inducible factor 1 subunit alpha (HIF-1α) was also observed in cisplatin-resistant ovarian cancer cells [60]. Further, in the xenograft model of non-small cell lung cancer, mice fed with GSPE show reduced VEGF levels [61].

#### 3.1.2. Apoptosis Promotions of Cancers

Bax and Bcl-2 are two Bcl-2 family members and usually form heterodimers to promote apoptosis [62]. The ratio of Bax/Bcl-2 determines the trend of apoptosis. It was observed that GSPE elevated the ratio of Bax/Bcl-2 in tongue squamous cell carcinoma to trigger apoptosis [63,64]. A similar induction effect was also found in breast cancer cell lines treated with PAs [65,66]. The higher ratio of Bax/Bcl-2 increased the expression of cleaved caspase-3 in cancer cells and resulted in the arrest in the G2/M phase cell cycle and the loss of MMP (matrix metalloproteinase). In addition to regulating the Bcl-2 family members, the proanthocyanidin polymer-rich fraction initiated apoptosis of cervical cancer cell lines through oxidative stress and mitochondrial damage [67]. Besides, proanthocyanidin B2 affected the glycolytic mechanism in HCC cells via suppressing the expression and nuclear translocation of pyruvate kinase M2 (PKM2), a key rate-limiting enzyme in glycolysis [68]. Then HIF-1α was released from the interaction complex PKM2/HSP90/HIF-1α and subsequently promoted apoptosis. These pieces of evidence suggested that apoptosis induction was a significant pathway for PAs against cancers.

#### 3.1.3. Cell Cycle Interference

Interfere with the cell cycle is also a key target in cancer therapy. The activity of cyclins and cyclin-dependent kinases (CDK) plays a controlling role during the G1 to S phase transition [69]. GSPE downregulated the expression of CDK2, CDK4, CDK6, and cyclins D1, D2, and E to cause a G0/G1 phase arrest. This blockade reduced proliferation and induced apoptosis of head and neck squamous cell carcinoma (HNSCC) cells [63,70]. Furthermore, treatment with GSPE also resulted in the level restoration of tumor-suppressor proteins such as Cip1/P21 and Kip1/P27, the CDK inhibitors [70].

#### 3.1.4. Immunosuppression

An increase in IL-10 and a decrease in IL-12 had been shown in the skin and draining lymph nodes of mice irradiated by ultraviolet (UV) [71]. Afterwards, antigen-presenting cells suffered in DNA damage and migrated to lymph nodes where the effector T cells subsequently exerted an alteration. Treatment with dietary GSPE enhanced the immune sensitivity and thus reversed UV radiation’s immunosuppressive effects in skin cancers [71,72,73].

#### 3.1.5. PI3K/AKT Pathway

Various growth factors, hormones, and cytokines activate the lipid kinase PI3K by binding to their homologous receptors, and then generate PIP3 on the plasma membrane, thus activating Akt [74]. Activated Akt directly phosphorylates many downstream substrates, including mTOR, to promote cell growth and protein synthesis in G1 cell cycle progression [75]. Studies showed that OPCs suppressed the PI3K/AKT/mTOR pathway to induce the death of neuroblastoma cells and gastric cancer cells [76,77]. NF-κB, a heterodimer consisting of two subunits, p65 and p50, is inactive in the cytoplasm due to the binding inhibition of nuclear factor κB kinase (IκB) [78]. When phosphorylated by Akt or IKK, IκB undergoes ubiquitination and degradation, then frees NF-κB. NF-κB thus enters the nucleus to activate the expression of genes closely related to cell survival and apoptosis, including Bcl-2 and MMP9. The expression levels of both AKT, IKK, and NF-κB have been found elevated in tongue squamous cell carcinoma [63]. Treatment with GSPE inhibited the phosphorylation of Akt and IKK and the translocation of NF-κB into the nucleus in Tca8113 cells.

#### 3.1.6. Epigenetic Modification

Epigenetic modification, such as DNA methylation and histone modification, is a heritable modification of gene expression that maintains the integrity of the actual DNA sequence but leads to cancer development [79]. Mudit et al. demonstrated that in in vitro analysis, GSPE played a role of anticancer in epigenetic mediating, synergistically and significantly reducing the activities of DNA methyltransferase (DNMT) and histone deacetylase (HDAC) in the skin cancer cell lines [80]. Similar modifications were discovered in proanthocyanidin-treated breast cancer cell lines [81,82].

#### 3.1.7. Self-Renewal Associated Pathway

Cancer stem cells (CSCs) have a vital function for formation, metastasis, and chemotherapy resistance of cancer by perpetual self-renewal. In the Hippo pathway, a self-renewal associated pathway, yes associated transcriptional regulator (YAP) and tafazzin (TAZ) control self-renewal and proliferation of cancer stem cells and reprogram nonstem cancer cells into cancer stem cells [83]. Shusuke et al. showed that OPCs lowered the level of YAP and TAZ in colorectal cancer cells [64]. Wnt/β-catenin signaling pathway is another pathway related to the self-renewal of CSCs. Without Wnt signaling, β-catenin was inhibited by synergistic phosphorylation of CK1 and APC/Axin/Glycogen synthase kinase 3β (GSK-3β) complex [84]. When an extracellular signal activated Wnt ligand, GSK-3β was released from the complex, causing β-catenin to be transported to the nucleus and activate Wnt target genes. In ovarian cancer stem cells, PAs up-regulated the phosphorylation of GSK-3β, which inhibited the activity of β-catenin and further decreased the expression of c-Myc and cyclin D1 [85]. The inhibitory effect of dietary GSPE had also been proved in melanoma, a highly malignant and aggressive form of skin cancer. In addition to promoting apoptosis and inhibiting proliferating cell nuclear antigen (PCNA) in human melanoma cells, GSPE decreased β-catenin levels in the cytoplasm and nucleus of the Wnt/β-catenin signaling pathway [37].

#### 3.1.8. Other Functions for Cancers

GSPE also exerted anticancer effects with anti-inflammatory properties. A clinical trial showed that GSPE inhibited and down-regulated COX-2/PGE2 pathways and significantly increased PGI2 and 15-HETE against lung cancer [33]. Dietary GSPE was also shown to target IGFBP-3 in lung cancer, and the specific mechanism may be related to the IGF-I/IGF-I receptor/IGFBP-3/phosphatidylinositol 3-kinase pathway [61]. Moreover, there is evidence that GSPE could also induce significant autophagy in HepG2 cells by enhancing the phosphorylation of p-JNK, p-ERK, and p-p38 MAPK while downregulating the expression of survivin [86]. Although the anticancer effects of PAs mainly focus on these mechanisms, more targets are worth exploring.

### 3.2. Cardiovascular Disease

#### 3.2.1. Cardiac Damage

PAs’ function in cardiac damage is mainly in preventing oxidative stress after ischemia repercussion injury and myocardial infarction (Figure 2). Pretreating PAs are claimed to limit injury or apoptosis of myocardial cells under different pathways, including P53/Bax, Bcl-2/Caspase 3 pathway, pancreatic eIF-2α kinase (PERK)/eukaryotic initiation factor 2α (eIF-2α) pathway and some critical factors involved in inflammation processes like reactive oxygen species (ROS), and tumor necrosis factor-alpha (TNF-α) production [87,88,89]. Rathinavel A. had also demonstrated that extracellular matrix (ECM) protein changes by PAs via BMP-4 could cope with oxidative damages after heart infarction and subsequent cardiac fibrosis [90]. It could also tackle chronic inflammation such as cell infiltration and high blood pressure caused by high carbohydrates and high-fat diet in rats [90]. Ischemic injuries related to free radical oxygen are attenuated by inhibiting gene expression of ASK1, nuclear factor-κB (NF-κB) and Cyclooxygenase-2 (COX-2) [91,92], increasing superoxide dismutase (SOD) activities and reducing the malondialdehyde (MDA) level [93,94]. All these unassuming advantages make a massive difference to the patient’s prognosis.

#### 3.2.2. Vascular Disease

Vascular health is essential in modern society as diseases like high blood pressure, atherosclerosis and derivatives have caused many deaths in recent years. Using medicine to manage a vascular problem is a regular operation for patients with a blood pressure problem. However, it appears to be practical to use dietary therapy to solve the problem. Substances like PAs are a choice for them. According to the experiment done by Odai.T et al., mean systolic pressure (MSP), mean diastolic pressure (MDP) and several vascular elasticity indexes are all improved by using PAs for three months [35]. Pretreatment with PAs before subarachnoid hemorrhage can relax the basilar artery, thus negating the effects of vacuolization, necrosis and apoptosis of endothelial cells in the brain after stroke [95,96]. Thrombosis occurrence is also inhibited by using GSPE under affecting thrombosis-promoting factors, for instance, P-selectin, von Willebrand factor (vWF), CAMs, demoting factors CD34, vascular endothelial growth factor receptor-2 (VEGFR2), and a disintegrin and metalloproteinase with a thrombospondin type one motif, member 13 (ADAMTS13) [97]. These findings mentioned above are realms of possibility for the use of PAs as a technique for treatment. Here, the PA benefits for vascular diseases are divided into three parts: prevention of atherosclerosis, blood pressure management, and administration of vascular remodeling and abnormal proliferation.

##### Prevention of Atherosclerosis

The whole development process of atherosclerosis is quite significant: leukocyte recruitment induced by vascular cell adhesion molecule-1 (VCAM-1) under a high-fat environment, foam cell formation, and collagen degradation in blood vessels [98]. As we knew, like the process, the function in PAs facing atherosclerosis is targeting one by one. PAs first restrain the oxidative process by the Nrf 2 pathway [99]. Cao et al. proved that GSPE could limit the asymptomatic carotid plaques or abnormal plaque-free carotid intima-media thickness (CIMT) and decrease mean maximum CIMT (MMCIMT), which can quell the arterial revascularization procedure [100]. Foam cell formation can also initiate atherosclerosis. PAs suppressed the expression of ACAT-1 in THP-1 cell by miRNA-9 level [101]. It even induced foam cells undergoing autophagy flux by impeding the expression of miR-96 with its downstream process Forkhead Box O1 (FOXO1)/mTOR)/p-mTOR/Autophagy marker Light Chain 3A/B (LC3A/B) or enhanced the ATP-binding cassette transporter-1 (ABCA1) and ATP Binding Cassette Subfamily G Member 1(ABCG1) protein level [102,103]. This precise suppression effectively halts the development of atherosclerosis.

##### Blood Pressure Management

The conventional drugs treating high blood pressure mainly comprise four parts, diuretic, β receptor antagonist, renin-angiotensin system (RAS) inhibitor and calcium channel blocker. Coincidentally, PAs are likewise effective as calcium channel blockers when applied to manage blood pressure. Scientists used Campomanesia xanthocarpa leaf extract (leaves abundant with PAs) to test whether it was possible to change ion channel activities. They eventually found that it caused the activation of ATP-sensitive potassium channels (KATP) and the reverse effect of l-type voltage-dependent Ca^2+^ channels [104]. It might also include myoendothelial gap junction signaling and calcium mobilization inside the cell during the process [105]. Large-conductance Ca^2+^-activated K^+^ channels (BKCa), voltage-gated K^+^(VK) channels are also responsible for vascular management by endothelium-dependent relaxation of arteries when using GSPE [106]. Nifedipine (NFD, an l-type calcium channel blocker), glibenclamide (GLB, an ATP-sensitive potassium channel blocker) and losartan (LOS, an angiotensin II type 1 receptor antagonist), for instance, are antagonists applied to rats or mice to confirm the channel-affecting result in the above studies mentioned [104,105,106]. Calcium and potassium channels are dedicated more through all ion channels in vascular regulating circumstances.

##### Administration of Vascular Remodeling and Abnormal Proliferation

Vascular remodeling is a secondary response after the oxidative stress process of blood vessels. As the antioxidant effect mentioned above, PA-induced oxidative response and inflammation factors are contracted to hinder abnormal proliferation and remodeling of endothelial cells. In rats with hypertension, PAs suppressed elevated TGF-β1 in small arteries with downregulation of the subsequent intercellular and intracellular cascade [107]. Upregulation of nitrate oxide (NO) level, downregulation of ROS, endothelin 1(ET-1), NADPH oxidase and some inflammatory factors like IL-1β, IL-6 and TNF-α also launch the effect [108,109]. Signaling pathways such as Janus Kinase 2 (Jak-2)/Signal transducer and activator of transcription 3 (STAT-3)/Cytosolic phospholipases A2 (cPLA2), ERK1/2, JNK1/2, and PI3K/Akt/GSK3β as well as the NF-κB pathway are also involved in impeding vascular smooth muscle cell (VSMC) proliferation [110,111]. The significant discovery of these findings is that pregnant mice with hypertension can also benefit from these mechanisms against oxidative effects without malformation, which gives us immense confidence to use PAs for dietary treatment [112].

### 3.3. Lipid Metabolism

Three powerful nutrient metabolic processes are essential in human beings, and they are all related to cell routine. Lipid metabolism is one of them with connections with several metabolic syndromes, including low-density lipoprotein (LDL) cholesterol, high-density lipoprotein (HDL) cholesterol and visceral adiposity [113,114,115]. Thus, commanding lipid metabolism is a big task that must be tackled. GSPE had been proved effective in losing weight, especially primarily type B2, which can prevent hypertriglyceridemia [114,115]. The weight loss is independent of genetic and environmental elements, with a common share that lowered triglyceride, total cholesterol, LDL and increased HDL [103,116,117]. Here, the detail of lipid metabolism is separated into three parts: gene regulation by PAs, prevention of lipid peroxidation and adjustment of lipid catabolic process.

#### 3.3.1. Gene Regulation by PAs

The gene is the basic unit controlling cell cycle and activities that can change the whole cell function and affect progeny cells when it has been altered. Therefore, scientists are always finding ways to treat disease from a genetic basis, with a long-lasting advantage. PAs have been found that can alter several key genes related to lipid hemostasis at transcription level and their mRNA expression, covering the mRNA expression of sterol regulatory-element binding proteins 1c (SREBP1c) and LD proteins, PPAR-α, miR-33a & miR-122 liver expression, ATP-binding cassette A1 mRNA and ABCA1 in vitro experiments [118,119,120,121]. In vivo, some genes such as enzyme acetyl-CoA acetyltransferase 2 (Acat2) for cholesterol metabolism or enzyme acyl-CoA synthetase long-chain family member 1 (Acsl1) for fatty acid ligation are manipulated by GSPE [121]. This gene regulation depends much on lipid metabolism.

#### 3.3.2. Prevention of Lipid Peroxidation

Lipid metabolism deregulation is also linked with oxidative stress (Figure 2). The use of oligomeric PAs with pterostilbene and niacin in 50:30:20 ratio shows a significant effect on the LDL/HDL ratio and atherogenic index suppression [122]. Its success in reducing LDL level with respect to increased HDL level is regarded as antilipid peroxidation. Restricted expression of cytochrome C and caspases-3 with activation of cardiac enzymatic and nonenzymatic antioxidant defense systems lead to less injury caused by hypercholesterolemia [123]. Regulating de novo lipid synthesis of HepG2 cells by suppressing ROS, GSH level and MDA production and the activation of antioxidant enzyme including glutathione peroxidase (GPx), catalase (CAT), and superoxide dismutase (SOD), can protect cells from damage [24,124]. Another experiment done by Yin.et al. had also shown activation of antioxidative enzymes such as SOD and GST, or restoration of lipid regulatory enzyme-like AMPK and ACC phosphorylation, to ameliorate lipid peroxidation damage [125].

#### 3.3.3. Adjustment of Lipid Catabolic Process

Anabolism and catabolism are always in equilibrium to sustain the balance of the process. Under high intake of fats, PAs try to induce catabolic activities by regulating enzyme activity, mainly enhancing cholesterol degradation and excretion. Sommella E discovered that Annurca polyphenolic extract (AAE), a kind of apple with abundant PAs, limits cholesterol production by diverting necessary materials citrate and acetyl-CoA to Kreb’s cycle [126]. The whole process is based on the reprogramming of hepatic cell metabolism and promoting mitochondrial respiration instead. In contrast, another team declared a different enzyme upregulation, LDLr and HNF-4α, with similar therapeutic effects as atorvastatin [127]. It is quite apparent that many of them were aiming to convert cholesterol into bile acid for diminution of cholesterol in plasma, although scientists got a different insight into the mechanism. On the other hand, GSPE upregulates the mRNA expression of 3-hydroxy-3-methylglutaryl coenzyme A reductase. Not only the protein levels of cholesterol-7α-hydroxylase (CYP7A1) but also mRNA CYP7A1 are improved when using GSPE, which is a key enzyme involved in bile acid (BA) synthesis [128]. Such differences might be achieved by routine administration, under the enhancement of BA production.

## 4. Toxicity Recovery and Synergic Effect of the Medicine

Little is known about PAs’ role as antidote for heavy metal poisoning, drugs’ toxicity or harmful substances in cigarettes. Heavy metals like lead and cadmium can cause heart damage through endoplasmic reticulum stress. GSPE either reduces lead accumulation in the heart with obstruction of lead-caused apoptosis or strikingly decreases those side effects caused by lead accumulation, such as anemia, liver dysfunction and DNA damage. Both effects are based on hindering RNA-like endoplasmic reticulum (ER) kinase/eukaryotic initiation factor 2α signaling pathway/Akt/glycogen synthase kinase 3 β phosphorylation, and miRNA153 and Akt/glycogen synthase kinase 3 β/Fyn-mediated Nrf2 activation, respectively [129,130]. Pretreating with PAs can offset cardio-oxidative stress markers by improving mitochondrial and respiratory chain enzyme activities.

PAs protect vascular health from exposure to nicotine and other substances in the cigarette. Black soybean seed coat extract (BSSCE) with a tremendous amount of PAs upregulate MMP-2 to prevent thinning of the vascular wall caused by the degradation of elastin and collagen fiber [131]. Remodeling pulmonary arteries are restricted by limiting pulmonary artery smooth muscle cell (PASMCs) proliferation by inhibition of the PPAR-γ/COX-2 pathway [132].

Clinical drugs may also harm our body, sometimes it can be avoided by changing to other drugs, sometimes it cannot because it is the appropriate treatment for the disease. PA use as a dietary therapy may help in this situation. For instance, chemical therapy medicine doxorubicin (DOX) is the usual drug for a tumor or immune disease, but it has a heart poisoning effect, which is not neglectable. PAs’ antioxidant effect can get used to it and prevent serum CK-MB and LDH activities as well as myocardial MDA and GSH contents and elevate serum catalase and myocardial SOD activities, as a result of protection of cardiac reduced glutathione (GSH) and management of total cardiac antioxidant (TAO) level [133,134]. Cardiotoxicity can also be prevented when using immunosuppressive drug cyclosporine A with GSPE, due to its antiapoptotic and antioxidant activities [135]. Aside from chemical therapies, additive titanium oxide nanoparticles TiO2 NPs also result in molecular damage. PAs can also alleviate this through the activation of nuclear factor erythroid 2 (NF-E2)-related factor 2 (Nrf2), NAD(P)H dehydrogenase[quinine] 1(NQO1), heme oxygenase 1 (HO-1) and glutamate-cysteine ligase catalytic subunit (GCLC) [135]. It might be a replenisher that could be used in clinics.

## 5. Conclusions and Prospects

Cancer, cardiovascular diseases and lipid/glucose metabolic disorders are major causes of mortality. Although we have the standard therapeutics for these diseases, patients still have to suffer from both the diseases and the treatments. Some flavonoids could be a solution to the dilemma as PAs are natural anticancer and health-care substances readily available from the diet. Therefore, it is regarded as one of the preferred drugs in many fields. Here we summarize the previous studies and consider that PAs should be used as a potential agent to treat cancer, cardiovascular diseases and lipid/glucose metabolic disorders by the possible mechanisms mentioned above. Some limitations still exist in current research. For example, a uniform standard for materials used in the research of PAs should be set up, which would serve to exclude interference from other substances. Besides, some in vitro studies in this review were designed without considering the PA bioavailability or metabolism. This might not fully represent the actual concentration of PAs in the corresponding human tissue. More systematic data of the distribution of PAs in human tissue and clinical trials are still needed. Therefore, we would like to see further work on the following aspects: PAs combined with standard therapy to improve the overall survival of cancer patients and reverse the anticancer drug resistance mechanisms; PAs helping to reduce the complications of cardiovascular diseases and metabolic syndrome, improving the quality of patients’ lives; investigation of the synergistic and antagonistic effects of combining the PAs and other drugs analyzed by the studies in vivo and in vitro, requiring further clinical trials.

## Figures and Tables

**Figure 1 molecules-25-05971-f001:**
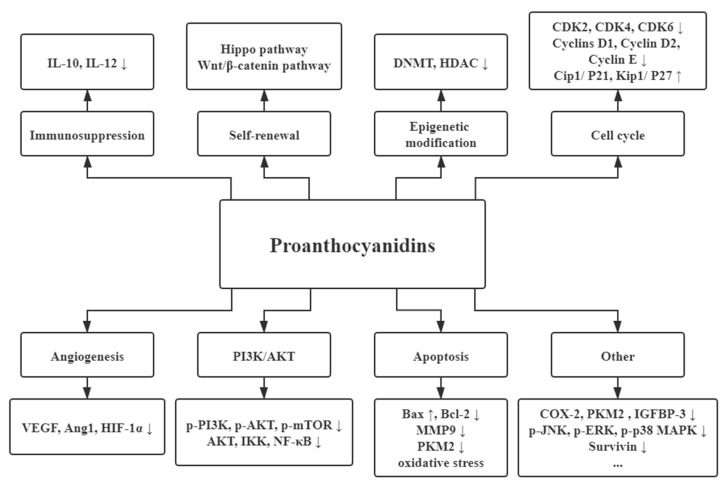
The dominating mechanisms of proanthocyanidins in cancer prevention or therapy. IL-10, interleukin-10; IL-12, interleukin-12; DNMT, DNA methyltransferase; HDAC, histone deacetylase; CDK2, cyclin dependent kinase 2; CDK4, cyclin dependent kinase 4; CDK6, cyclin dependent kinase 6; VEGF, vascular endothelial growth factor; Ang1, angiopoietin 1; HIF-1α, hypoxia inducible factor 1 subunit alpha; Bax, BCL2 associated X, apoptosis regulator; Bcl-2, BCL2 apoptosis regulator; MAPK, mitogen-activated protein kinase; MMP9, matrix metalloproteinase; PI3K, phosphoinositide-3-kinase; AKT, protein kinase B; p-mTOR, phosphorylated mechanistic target of rapamycin kinase; IKK, inhibitor of nuclear factor κB kinase; NF-κB, nuclear factor κB subunit 1; COX-2, cyclooxygenase-2; PKM2, pyruvate kinase M2; IGFBP-3, insulin like growth factor binding protein-3; JNK, Jun amino-terminal kinase; ERK, extracellular regulated protein kinase.

**Figure 2 molecules-25-05971-f002:**
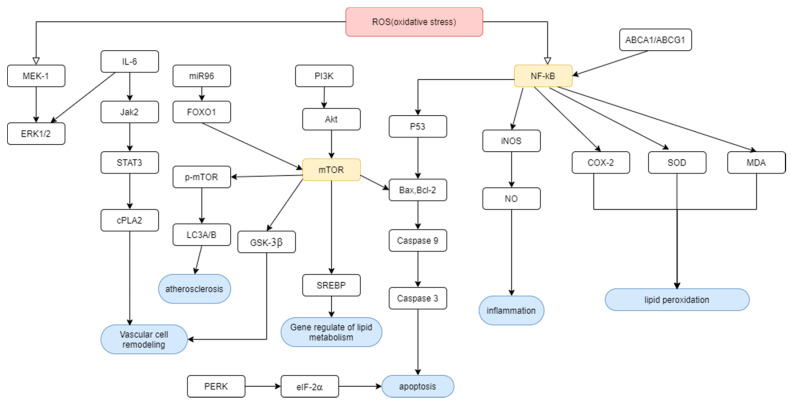
Grape seed proanthocyanidin extracts (GSPE) manipulate cardiovascular diseases and lipid metabolism caused by ROS under these genes or factors’ regulation. From left to right: ROS, reactive oxygen species; ERK1/2, extracellular signal regulated kinase 1/2; IL-6, interleukin-6; Jak2, janus kinase 2; STAT3, signal transducer and activator of transcription; cPLA2, cytosolic phospholipases A2; FOXO1, forkhead Box O1; mTOR/p-mTOR, mechanistic target of rapamycin kinase/phosphorylated mTOR; LC3A/B, autophagy marker light chain 3A/B; PI3K, phosphoinositide 3-kinases; Akt, protein kinase B; GSK-3β, glycogen synthase kinase 3β;SREBP, sterol regulatory element-binding proteins; NF-κB, nuclear factor-κB; Bax, BCL2 associated X, apoptosis regulator; Bcl-2, BCL2 apoptosis regulator; PERK, pancreatic eIF-2α kinase; eIF-2α, eukaryotic initiation factor 2α; iNOS, inducible nitric oxide synthase; NO, nitrate oxide; COX-2, cyclooxygenase-2; SOD, superoxide dismutase; MDA, malondialdehyde; ABCA1/ABCG1, ATP-binding cassette transporter-1/ATP binding cassette subfamily G member 1 (ABCA1/ABCG1).

**Table 1 molecules-25-05971-t001:** The basic structure of flavonoid, its main subclasses, and the majority of proanthocyanidins (PAs).

Compound	Basic Structure
Flavonoid	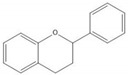
Flavonol	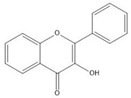
Flavone	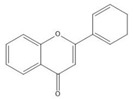
Flavan-3-ol	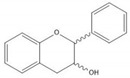
Anthocyanidin	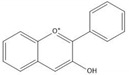
Flavanone	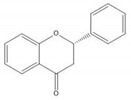
Isoflavone	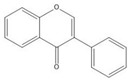
(+)-Catechin	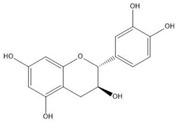
(−)-Epicatechin	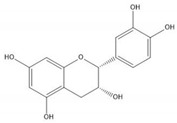
Proanthocyanidin A1	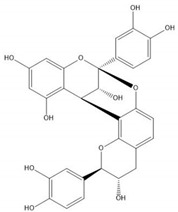
Proanthocyanidin A2	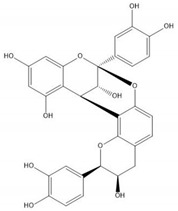
Proanthocyanidin B1	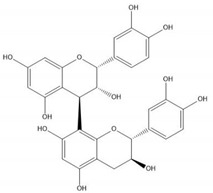
Proanthocyanidin B2	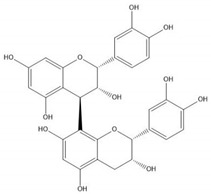
Proanthocyanidin B3	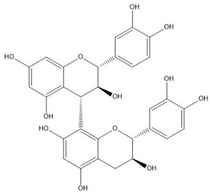
Proanthocyanidin B4	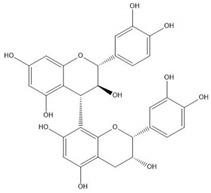
Proanthocyanidin B5	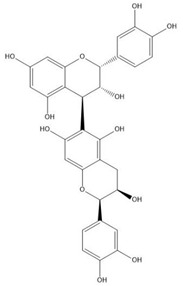
Proanthocyanidin B6	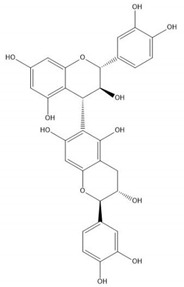
Proanthocyanidin B7	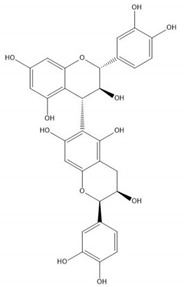
Proanthocyanidin B8	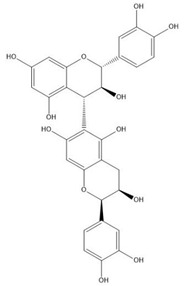

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
