# Peer review of "Proanthocyanidins Should Be a Candidate in the Treatment of Cancer, Cardiovascular Diseases and Lipid Metabolic Disorder"

_molecules, 2020, doi:10.3390/molecules25245971_

Round 1

Reviewer 1 Report

The study of the biological activity of flavonoids is one of the major challenges in the food science field. Over the last decade, proanthocyanidins are attracting attention due to their health benefits. This review includes a brief description of the structure of flavonoids (focusing mainly on proanthocyanidins) and the method of isolation of proanthocyanidins, as well as a summary of studies regarding the beneficial effects of proanthocyanidins against cardiovascular diseases, cancer and lipid metabolic disorders.

After reading this paper I have a number of major concerns, which should be addressed by the authors in case the handling editor invites resubmission of a revised version:

  • There is a lack of information about the metabolism and bioavailability of proanthocyanidins. This is critical to understand how proanthocyanidins exert their beneficial effects and to determine what molecules are responsible of the biological effects.
  • The anticancer activity of proanthocyanidins is mainly supported by in vitro studies. Some of them are not relevant from a physiological point of view. An example of this is reflected in the study by Wen et al. (reference 28) in which endothelial cells were exposed to a grape seed extract, what is not plausible from an in vivo point of view. More examples of this can be found through the manuscript.
  • In line with the previous point, there are an important number of relevant studies that have not been included.
  • Besides, the absence of clinical trials regarding the anticancer activity of proanthocyanidins reduces the quality of the review.

Author Response

Thank you for your suggestion. We provided new reference for explaining the general usage of PAs in complexed form and its bioavailability. Some questions are still cannot explain by existing researches. The following lines highlighted are those revisions we have been made. Hope to hear advices from you.

Ln 50    Revised the subtitle according to the content.

Ln 52-80    1. Rewrote and summarized the isolation steps of proanthocyanidins (PAs), demonstrating some technologies that are widely used for isolation. 2. Added the information about the metabolism and bioavailability of PAs. 3.Rewrote the sentence.

Ln 82-138    Additional lines are added to describe: 1. why PAs are possible therapeutic supplement and its importance. 2. Why complex PAs instead of pure PAs are used in clinical trails and experiments. 3. Clinical trails in 10 years, that testing possible functions, are provided. 4. Better options of mixed PAs, like GSPE and LP, and the underlying reasons that it’s more likely to be used in clinical situation. 5. General systematic response are triggered by PAs rather than direct actions.

Few questions that are not able to covered in current research 1. wheather specific PAs act whilst others do not or individual PAs in a preparation act additively or synergistically to give a protective action.  2. Actual compositions of widely tested preparations, such as grape seed proanthocyanins extract (GSPE), cannot be provided as it variate in different tablets as there are no standard, by only knowing there are about 40-90% of PAs contained. 3. Could prolonged consumption impair or limit a host’s ability to respond to an environmental or microbial challenge? There is no experimental data to answer the questions.

  1. Summarized the current situation of in vitro and in vivo studies of PAs, and PAs' components used as the intervention. 2. The number and the gap of completed clinical trials on PAs. 3. Rewrote the sentence.

Ln 144-146    PAs' anti-cancer activities stay in the stage of in vitro and in vivo studies.

Ln 202    In vitro studies.

Ln 223-224    A clinical trial.

Reviewer 2 Report

The authors have reviewed the physical and biological properties of proanthocyanins (PAs), their putative antioxidant, anti-inflammation properties, and influences on signal pathways that are important in many diseases.  They conclude that PAs should be a candidate in the treatment of cancer, cardiovascular diseases, and lipid metabolic disorder.

This is a thorough review of the putative health-promoting properties of proanthocyanins. However, while there is a discussion of the isolation and characterization of individual PAs, most of the health-associated studies are done with PA-enriched mixtures or extracts. This leaves the major question as to whether all PAs are equally effective against underlying causes of disease, specific PAs act whilst others do not or individual PAs in a preparation act additively or synergistically to give a protective action. While advocating a general consumption of PAs may be no bad thing, a more detailed knowledge of mechanisms of action, alone or in combination, of PAs and their reproducibility and robustness in a clinical situation is required before they can be advocated for treatment or ancillary management of severe diseases.

The authors should summarise and highlight from available information on what clinical trials have been done or are ongoing with PA preparations. What clinical or pre-clinical studies have been reported with pure PAs? Are there any indications that the actions of PAs are additive or synergistic? Given that many studies are done with mixed extracts of PAs, an indication of the likely compositions of widely tested preparations, such as grape seed proanthocyanins extract (GSPE), should be given.

PAs appear to have a wide range of actions on important metabolic pathways. To what extent can these be ascribed to the targeted and direct action of PAs on parts of these response pathways or are they just down-stream consequences of a more general systemic response to PAs. Given that PAs seem to have beneficial effects this may seem trivial, but a clear understanding of the likely mode/s of action and replicability is required before they should be recommended for widespread use against disease.

PAs appear to modulate many metabolic systems involved in host responses to disease and suppress or prevent potentially adverse changes. Is there any evidence that they can alter metabolism in a healthy individual? Could prolonged consumption impair or limit a host’s ability to respond to an environmental or microbial challenge?

Ln 63-64         What is the purpose of ultrasound? Does it increase the yield or change the conformation of PAs? How reproducible?

Ln 68-77         What proportions of data are from clinical, pre-clinical, or experimental studies? Are there any studies with pure PAs? Summarise the general compositions of extensively used PA-preparations?

Ln 144-147     In vitro or in vivo?  

Ln 225-236     To what extent has this been demonstrated in vivo and in a clinical situation?

Ln 261-270     To what extent is this information in vitro. What has been demonstrated in vivo?

Author Response

Thank you for your suggestion. We provided new reference for explaining the general usage of PAs in complexed form and its bioavailability. Some questions are still cannot explain by existing researches. The following lines highlighted are those revisions we have been made. Hope to hear advices from you

Ln 50    Revised the subtitle according to the content.

Ln 52-80    1. Rewrote and summarized the isolation steps of proanthocyanidins (PAs), demonstrating some technologies that are widely used for isolation. 2. Added the information about the metabolism and bioavailability of PAs. 3.Rewrote the sentence.

Ln 82-138    Additional lines are added to describe: 1. why PAs are possible therapeutic supplement and its importance. 2. Why complex PAs instead of pure PAs are used in clinical trails and experiments. 3. Clinical trails in 10 years, that testing possible functions, are provided. 4. Better options of mixed PAs, like GSPE and LP, and the underlying reasons that it’s more likely to be used in clinical situation. 5. General systematic response are triggered by PAs rather than direct actions.

Few questions that are not able to covered in current research 1. wheather specific PAs act whilst others do not or individual PAs in a preparation act additively or synergistically to give a protective action.  2. Actual compositions of widely tested preparations, such as grape seed proanthocyanins extract (GSPE), cannot be provided as it variate in different tablets as there are no standard, by only knowing there are about 40-90% of PAs contained. 3. Could prolonged consumption impair or limit a host’s ability to respond to an environmental or microbial challenge? There is no experimental data to answer the questions.

1.Summarized the current situation of in vitro and in vivo studies of PAs, and PAs' components used as the intervention. 2. The number and the gap of completed clinical trials on PAs. 3. Rewrote the sentence.

Ln 144-146    PAs' anti-cancer activities stay in the stage of in vitro and in vivo studies.

Ln 152    Changed abbreviation.

Ln 202    In vitro studies.

Ln 223-224    A clinical trial.

Ln 292-293 297-301    Additional lines are added to provide in vivo evidence by using ion channel blockers (Nifedipine , glibenclamide, osartan) with experimental rats or mouse.

Ln 333-334    Additional words are added to clarify in vitro and in vivo experiment. It is still hard to figure out the extent it take under current experimental data.

Round 2

Reviewer 1 Report

The authors have included some information about metabolism and bioavailability. However, a much deeper description must be performed. The authors indicate that “oligomers were preferentially absorbed for glucuronidation”. What oligomers? What are the main glucuronides detected in the bloodstream? What range of concentrations are detected in the bloodstream? Are these molecules detected in different tissues? If so, at what concentrations? Are sulfate and/or methyl esters of proanthocyanidins detected? If so, what concentrations? Can these metabolites reach systemic tissues? If so, what concentrations? This information is essential, since it sets the basis to correctly design physiological relevant in vitro studies. Additionally, it allows to determine what determine from a critical point of view what in vitro studies provide valuable information to consider proanthocyanidins “good” candidates for the treatment of chronic diseases.

The authors have included relevant clinical trials mainly related to cardiovascular health and lipid metabolism. Due to the relevance of cancer in this review, I suggest to include information recently reviewed about the anticancer effects against breast cancer exerted by proanthocyanidins (PMDI: 32784973).

The title of the manuscript is not in concordance with the studies described in this review. This is mainly reflected in the lines 158-161 where the authors indicate that “studies of anti-carcinogenic activities only stay in the exploration stage of mechanism in vitro and in vivo, and clinical trials for cancer therapy obtained few results with predominant clinical significance or were still in progress”. According to this statement, we understand the molecular mechanisms of action of proanthocyanidins far less than is necessary to design clinical trials that will prove a therapeutic effect of these molecules. Considering this, the title of this manuscript fails to reflect the current evidence and has to be modified.

Most of the in vitro studies included in this review are designed without considering the metabolism and bioavailability of proanthocyanidins. This reduces their physiological relevance. This point must be criticized in the conclusions as a limiting factor to consider proanthocyanidins “good” candidates for the treatment of chronic diseases.

Author Response

Thank you for your further review comments, we will ask the questions in attachment below.

Reviewer 2 Report

The authors have dealt adequately with all of the queries raised

Author Response

Thank you very much for your comments.